# High-Accuracy Spectral Measurement of Stimulated-Brillouin-Scattering Lidar Based on Hessian Matrix and Steger Algorithm

Zhiqiang Liu [1], Jie Sun [1], Xianda Zhang [1], Zhi Zeng [1], Yupeng Xu [1], Ningning Luo [1], Xingdao He [1,2,3] and Jiulin Shi [1,2,3,*]

1    Jiangxi Provincial Key Laboratory of Opto-Electronic Information Science and Technology, Nanchang Hangkong University, Nanchang 330063, China
2    Key Laboratory of Nondestructive Test (Ministry of Education), Nanchang Hangkong University, Nanchang 330063, China
3    National Engineering Laboratory for Nondestructive testing and Optoelectric Sensing Technology and Application, Nanchang Hangkong University, Nanchang 330063, China
*    Correspondence: jiulinshi@126.com

**Abstract:** The measurement accuracy of Brillouin scattering spectra is crucial for ocean remote sensing by Brillouin scattering lidar. Due to the limited resolution of ICCD cameras, the traditional processing methods remain at the pixel or partial sub-pixel level, which cannot meet the requirements of high-performance lidar. In this paper, to extract the frequency shift with high precision from stimulated Brillouin scattering (SBS) lidar, a novel spectral processing method with sub-pixel recognition accuracy is proposed based on the Hessian matrix and Steger algorithm combined with the least square fitting method. Firstly, the Hessian matrix and Frangi filter are used for signal denoising. Then, the center points of SBS spectra at the sub-pixel level are extracted using the Steger algorithm and are connected and classified according to the signal type. On that basis, the frequency shifts of Brillouin scattering are calculated by using the center and radii of interference spectra after through fitting by the least squares method. Finally, the water temperatures are inverted by using the frequency shifts of Brillouin scattering. The results show that the processing method proposed in this paper can accurately calculate the frequency shift of Brillouin scattering. The measured errors of frequency shift are generally at an order of MHz, and the inversion accuracy of water temperature can be as low as 0.14 °C. This work is essential to the application for remote sensing the seawater parameters by using the Brillouin lidar technique.

**Keywords:** Brillouin lidar; spectral measurement; Hessian matrix; Steger algorithm

## 1. Introduction

As an essential object of ocean exploration, physical parameters such as temperature, salinity, and sound speed of seawater are of great significance to be measured in real time and accurately in oceanography. Currently, the ocean surface parameters can be obtained by satellites, while the distribution profiles underwater are extracted by using conductivity-temperature-depth (CTD) instruments, buoys, or gliders [1–4]. However, these techniques do not allow rapid, accurate, and real-time range-resolved monitoring. Therefore, a flexible, cost-efficient, and real-time remote sensing technique is highly desirable. As an alternative approach, Brillouin-scattering-based lidar provides a promising solution. It can invert the physical parameters by measuring the frequency shift and linewidth of the Brillouin spectrum in seawater [5–7]. Among them, SBS lidar based on Fabry-Perot (F-P) etalon and intensified charge-coupled device (ICCD) can measure the physical parameters of a certain point in the seawater in a short time [8–11].

In ocean remote sensing by using SBS lidar, the retrieval precision of seawater parameters is affected by the spectral measurement accuracy of Brillouin scattering. Based

on the interference imaging principle of F-P etalon, the received spectra will be divided into two different frequency components: Rayleigh scattering and Brillouin scattering, and then are collected and presented by an ICCD camera in the form of two-dimensional ring interference spectra. Several methods have been proposed in previous works to process two-dimensional ring interference spectra to obtain the frequency shift and linewidth of Brillouin scattering, such as the direct-reading method [8,12], circle-to-line interferometer optical (CLIO) system [13], cylindrical lens compression method [14], data-fold method [15–17], and so on. Since there is no noise reduction, the processing accuracy of the direct-reading method can only reach the pixel level. The processing accuracy of the cylindrical lens compression method and CLIO system is greatly limited by the resolution of ICCD. The data folding method transforms the two-dimensional interferogram into one-dimensional line-type spectrogram after spectrum processing, its accuracy is only pixel level. For the resolution of 1024 × 256 pixels of ICCD, a pixel-level deviation could bring a frequency shift deviation of tens of MHz, and the corresponding temperature inversion deviation could reach the degree of Celsius [18]. Therefore, it is necessary to further improve the spectral processing accuracy for obtaining high inversion accuracy of seawater parameters by using SBS lidar.

The critical process to extract the frequency shift with high accuracy from the Brillouin scattering spectra depends on the recognition of ring-shaped interference spectra formed by F-P etalon. Regarding the detection of circles in spectra, the Hough transform circle detection (HTCD) algorithm is a common detection method [19]. The traditional HTCD method accumulates the votes by enumerating the combination of parameter values., and it takes a huge amount of computation and memory consumption [20]. Although some improved HTCD algorithms have been proposed successively, such as the random Hough transform circle detection (RHTCD) [21], gradient Hough transform circle detection (GHTCD) [22], etc., these algorithms need to be known the radius search range in advance, which complicates the processing process. On the other hand, the HTCD method also has a false detection rate, and the accuracy can only reach the pixel level.

The purpose of the present work is to develop a novel processing method based on the Hessian matrix and Steger algorithm for obtaining more accurate spectral information of SBS lidar and reducing the operating complexity of signal processing. This paper is organized as follows. Firstly, the structure and working principle of SBS lidar are briefly introduced, and Brillouin scattering spectra of seawater at different temperatures are measured by using SBS lidar. Then the principle of spectral information extraction and classification based on the Hessian matrix and Steger algorithm is presented. Based on the spectral information extraction and classification method, the frequency shift of Brillouin scattering is extracted through spectrum denoising, interference ring positioning, and parameter fitting, respectively. Finally, the water temperature is inverted by using the extracted frequency shift. The work is expected to pave the way for the automatic recognition and classification of spectral signals of Brillouin lidar in ocean remote sensing.

## 2. Method

### 2.1. SBS Lidar System

Figure 1 shows the optical configuration of SBS lidar. The laser used in the measurement is a seed-injected Nd: YAG pulsed laser with an operating wavelength of 532 nm, pulse frequency of 10 Hz, pulse duration of 7 ns (full width at half maximum, FWHM), a divergence angle of 0.45 mrad, and a single-beam pulsed laser energy of 250 mJ. The linewidth of the single-longitudinal mode is 90 MHz by switching on a seed laser. A focusing system based on the Galileo telescope configuration was designed to regulate the focal length of laser beams in water. The distance between the concave and convex lenses of the Galileo telescope can be adjusted, which can not only flexibly change the position of the detection point but also enhance the transmission ability of the laser in water. The measurements were carried out by employing an adjustable seal chamber to obtain the different temperatures of seawater.

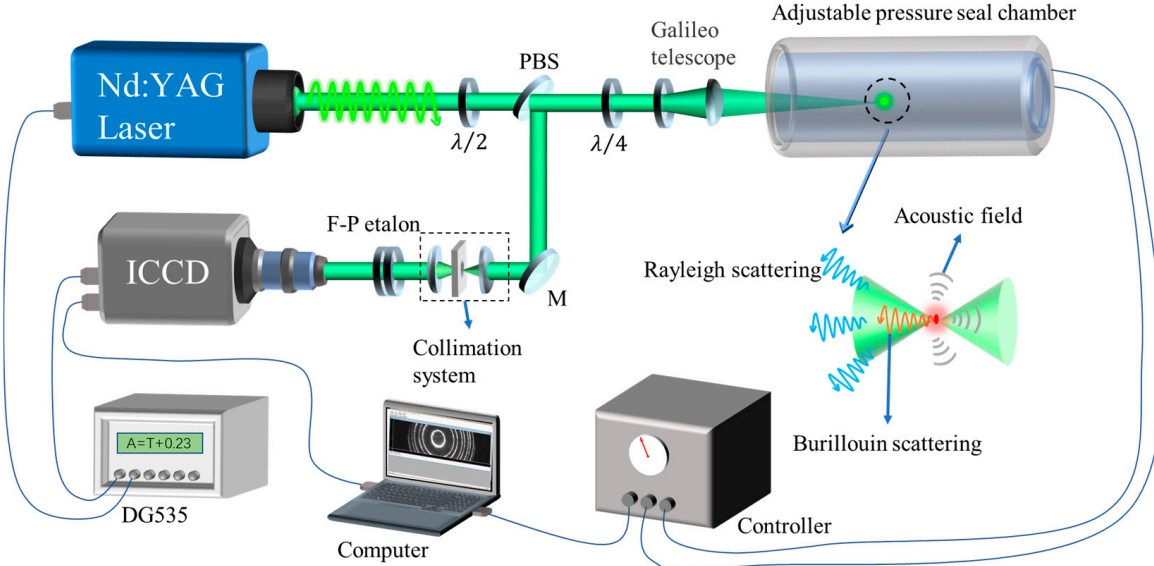

**Figure 1.** Optical setup of SBS lidar. λ/2 is a half-wave plate, λ/4 is a quarter-wave plate, and PBS is a polarization beam splitter.

As shown in Figure 1, the output laser beams from the laser with the vertical polarization are focused into seawater by the focusing system after passing through the λ/2 plate, PBS, and λ/4 plate in turn. The SBS signal is excited when the laser energy at the focal point reaches the threshold value. Then, the backward SBS signal passes through λ/4 plate and PBS and is reflected into the interferometer system consisting of F-P etalon and ICCD camera. The spectra can be obtained by using F-P etalon with a free spectral range (FSR) of 20.1 GHz and recorded by an ICCD camera. The resolution size of the ICCD is 1024 × 256, and the pixel size is 26 μm × 26 μm. The pulse delay time of the lidar system is accurately controlled by the time schedule controller (DG 535).

## 2.2. Brillouin Spectra Obtained from Interferometer System

The principle of the interferometer system based on the F-P etalon and ICCD camera is shown in Figure 2. The received Rayleigh and Brillouin scattering signals present in the focal plane after passing through the F-P etalon and focusing lens. Since the wavelength of Brillouin scattering light is shifted to a certain extent compared to Rayleigh scattering light, it will appear as a series of concentric double-ring structures with different diameters on the interference pattern.

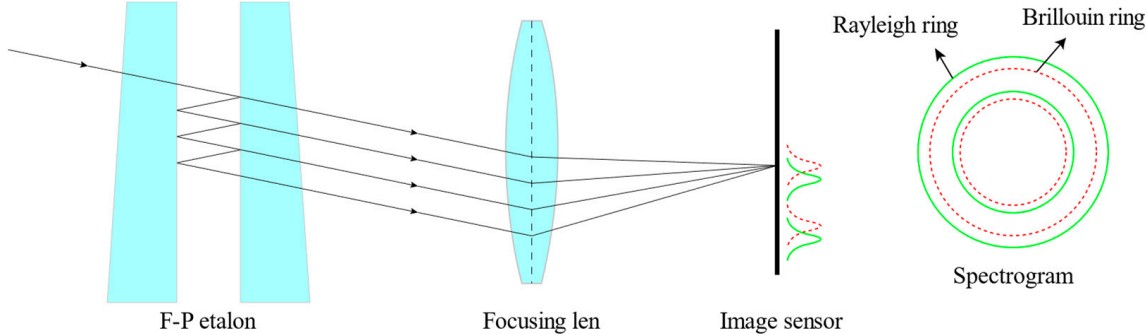

**Figure 2.** Principle of beam frequency division system based on the F-P etalon and ICCD camera.

The received signals are a group of isoclinic interference rings formed by the F-P etalon and ICCD camera. Figure 3 shows the measured scattering spectrum of seawater. Based on the interference spectrum, the frequency shift of Brillouin scattering can be calculated according to the information on the ring radius of adjacent stages.

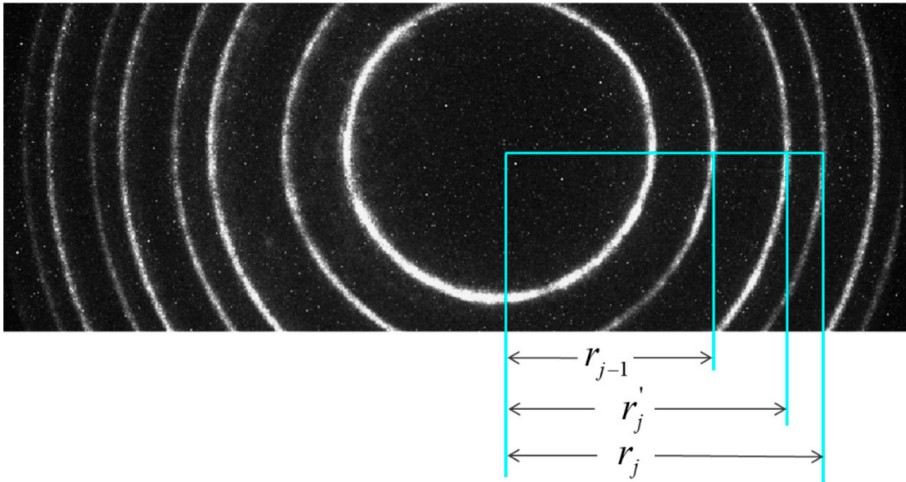

**Figure 3.** Measured scattering spectrum in seawater.

Here, $r_j$ and $r_{j-1}$ represent the ring radii of (j − 1)-th and j-th order of Rayleigh scattering signal, respectively, and $r'_j$ represents the ring radius of j-th order of Brillouin scattering signal. Then the frequency shift of Brillouin scattering $v_B$ can be calculated by using Equation (1):

$$v_B = \frac{r_j^2 - r_j'^2}{r_j^2 - r_{j-1}^2} \times FSR \tag{1}$$

*FSR* of Equation (1) is the free spectral range of F-P etalon. It can be seen from Equation (1) that when the FSR of the F-P etalon remains unchanged, the error of the calculated frequency shift is directly related to the ring's radii at all levels. Therefore, the accurate measurement of the ring's radii is extremely important for measuring the frequency shift of Brillouin scattering. During the signal acquisition process, the proportion of the rings in the scattering spectrum should be expanded as much as possible so that the frequency shift corresponding to each pixel in the adjacent rings is small to reduce the error brought by the measurement process of ring radius.

### 2.3. Proposed Method

The flow chart of the processing method proposed in this paper is shown in Figure 4. Firstly, to reduce the influence of noise in subsequent processing, the Frangi filter is used to denoise the spectrum. Secondly, to accurately find the center position of the bright rings, the Steger algorithm is used to identify the sub-pixel center of the rings. Lastly, the identified sub-pixel center points are fitted with the least squares method to obtain the radii of the rings so as to calculate the frequency shift of Brillouin scattering. The following chapters will present the principle of the method used in each part and show the specific performance of the experimental spectra after processing.

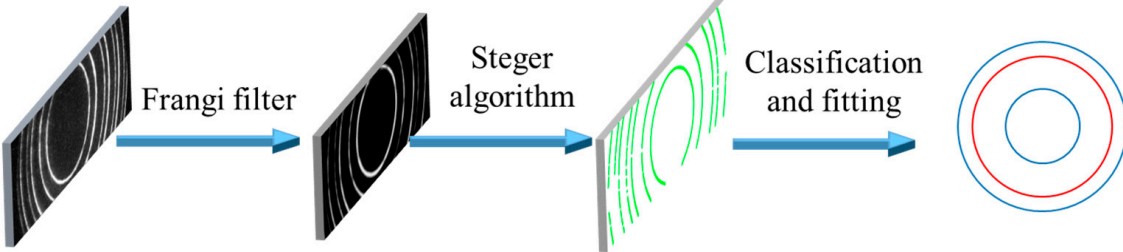

**Figure 4.** Schematic diagram of the spectral processing process.

2.3.1. Spectral Noise Removal Algorithm

A series of concentric rings in the spectrum can be viewed as a curve structure, and noise can be seen as bright spot structures. Both can be identified by using the Hessian matrix. The Hessian matrix is a square matrix of second-order partial derivatives of a multivariate function that describes the local curvature of the function. Let $I(x, y)$ represent the intensity value of the interference spectrum where the abscissa is $x$ and the ordinate is $y$. Then the Hessian matrix $H$ can be expressed as:

$$H = \begin{pmatrix} \frac{\partial I}{\partial^2 x} & \frac{\partial I}{\partial x \partial y} \\ \frac{\partial I}{\partial x \partial y} & \frac{\partial I}{\partial^2 y} \end{pmatrix} \tag{2}$$

The first and second order partial derivatives can be calculated based on the two-dimensional Gaussian function $g(x, y; \sigma)$, and then the Gaussian partial derivative convolution kernel can be obtained. The partial derivatives can be obtained by convolving the original spectrum with the Gaussian partial derivative convolution kernel. Therefore, Equation (2) can be rewritten as Equation (4). Here, $\sigma$ is a predefined Gaussian variance variable and specifies the size of the convolution kernel.

$$g(x, y; \sigma) = \frac{1}{2\pi\sigma^2} e^{-\frac{x^2 + y^2}{2\sigma^2}} \tag{3}$$

$$H = \begin{pmatrix} \frac{\partial I}{\partial^2 x} & \frac{\partial I}{\partial x \partial y} \\ \frac{\partial I}{\partial x \partial y} & \frac{\partial I}{\partial^2 y} \end{pmatrix} = \begin{pmatrix} I(x, y) \times g''_{xx}(x, y; \sigma) & I(x, y) \times g''_{xy}(x, y; \sigma) \\ I(x, y) \times g''_{xy}(x, y; \sigma) & I(x, y) \times g''_{yy}(x, y; \sigma) \end{pmatrix} \tag{4}$$

The eigenvalues of matrix $H$ are denoted as $\lambda_1$ and $\lambda_2$, and the corresponding eigenvectors are $e_1$ and $e_2$. The directions of $e_1$ and $e_2$ are the smallest and largest intensity values change at the point, respectively. Figure 5 shows the direction of the eigenvectors of the curve structure and the relative modulus length. The intensity represents the relative number of photons received by the ICCD camera. The values of $\lambda_1$ and $\lambda_2$ represent the change rates of intensity values in each direction. If there is a curve structure at a certain point, $e_1$ corresponds to the tangent direction, and the intensity value in this direction remains almost unchanged, so $\lambda_1 \approx 0$. On the contrary, $e_2$ corresponds to the normal direction, and the intensity value of this direction changes greatly, thus $|\lambda_2| \gg 0$. At the same time, the change rate of the intensity value of the point-like structure is relatively large along each direction, so the $|\lambda_1| \gg 0$ and $|\lambda_2| \gg 0$. Table 1 shows the relationships between different structures in the two-dimensional spectrum and the two eigenvalues of the Hessian matrix. So we can set the filtering conditions to enhance the ring structure and suppress the noise point.

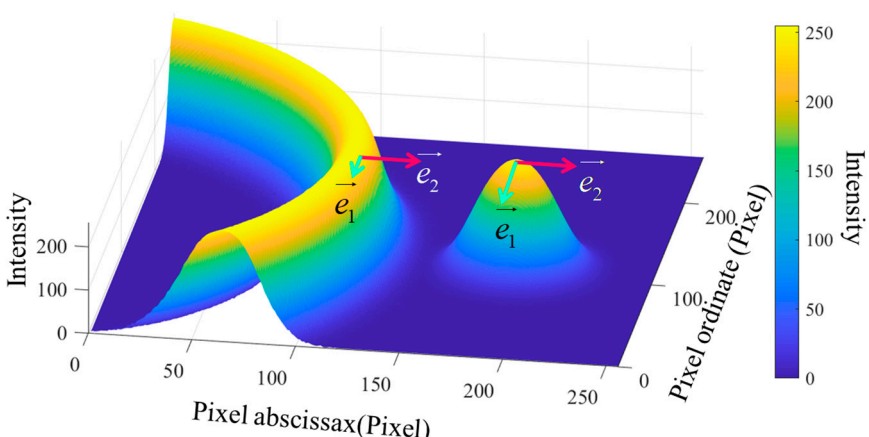

**Figure 5.** Schematic diagram of the eigenvectors and norm of the Hessian matrix (The curve structure is signal, and the point structure is noise.).

**Table 1.** The relationship between the eigenvalues and the structure types ("L" means close to $0(|L| < 10^{-4})$, "H−" means a large negative value(H− > 0.1), and "H+" means a large positive value(H+ > 0.1). The value range of the eigenvalues is calculated according to the experimental spectra.

| Structure Type | $\lambda_1$ | $\lambda_2$ |
| --- | --- | --- |
| bright fringe | L | H− |
| dark fringe | L | H+ |
| bright spot | H− | H− |
| dark spot | H+ | H+ |

Based on the above principle, Frangi et al. have formulated a multi-scale response function and targeted enhancement of the curve structure by traversing different variance values [23]. The response function of the 2-D curve structure can be expressed by Equation (5), where $\beta$ and $c$ are the set parameters that control the spot structure and the background output sensitivity, respectively. To enhance the curve structure of different thicknesses, the value of the variance $\sigma$ is selected as an interval range instead of a single value. The final output value $V_o(x, y)$ of a position is the maximum value of $V_o(\sigma)$, as shown in Equation (8). The variance range depends on the width of the rings, and the suggested variance range is [2,4] in our experimental spectra. Figure 6 shows the light intensity distribution of the experimental spectra after Frangi filter processing. It can be seen that the background noise and isolated noise points of spectra are significantly suppressed, and the interference rings (curve structure) are smoothed and enhanced to a certain extent.

$$V_o(\sigma) = \begin{cases} 0, if(\lambda_2 > 0) \\ \exp\left(-\frac{R_B^2}{2\beta^2}\right)\left[1 - \exp\left(-\frac{S^2}{2c^2}\right)\right] \end{cases} \tag{5}$$

$$R_B = \frac{\lambda_1}{\lambda_2} \tag{6}$$

$$S = \|H\|_F = \sqrt{\lambda_1^2 + \lambda_2^2} \tag{7}$$

$$V_o(x, y) = \max_{\sigma_{\min} < \sigma < \sigma_{\max}} V_o(x, y; \sigma) \tag{8}$$

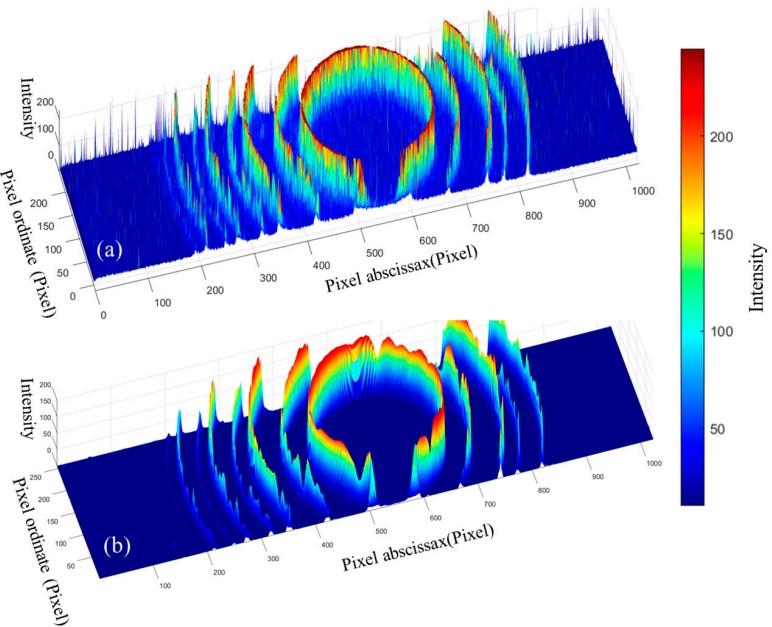

**Figure 6.** (**a**) Original spectrum of experimental measurement; (**b**) spectrum after noise removal using Frangi filter.

### 2.3.2. Extraction of the Centerline of Interference Rings

Considering that the interference rings have a certain width, the Steger algorithm is used to locate the center line of the sub-pixel of the rings [24]. Firstly, there is the proposed variance value in Equation (3), that is $\sigma \geq \omega/\sqrt{3}$. Here $\omega$ is the linewidth of the intensity profile of the interference ring. To make the center point presents a greater discrimination degree, it is necessary to set $\sigma = \omega/\sqrt{3}$. In this case, the absolute value of the second derivative is maximized at the center. In addition to the second-order partial derivative, the first-order partial derivative $N$ is also required to be calculated by using the same calculation method as used in the second-order derivative, which is obtained by convolving the image with the Gaussian first-order derivative convolution kernel, as shown in Equation (9). Secondly, the Hessian matrix is employed to construct the Taylor quadratic expansion polynomial of the intensity value at each point. Assume that $r_x$, $r_y$, $r_{xx}$, $r_{xy}$, $r_{yy}$ denote $\frac{\partial I}{\partial x}$, $\frac{\partial I}{\partial y}$, $\frac{\partial^2 I}{\partial x^2}$, $\frac{\partial^2 I}{\partial x \partial y}$, $\frac{\partial^2 I}{\partial y^2}$, respectively, then the calculation equation of $I_f$ is expressed as Equation (10). By calculating the extremum points of the polynomial, the position of the desired center point can be obtained.

$$N = \left( \frac{\partial I}{\partial x}, \frac{\partial I}{\partial y} \right) = \left( I(x,y) * g'_x(x,y;\sigma), I(x,y) * g'_y(x,y;\sigma) \right) \tag{9}$$

$$I_f(dx, dy) = I_0 + r_x dx + r_y dy + r_{xy} dx dy + r_{xx} dx^2 + r_{yy} dy^2 \tag{10}$$

To reduce the amount of calculation needed to search for extreme points in the binary space, the algorithm takes the search for extreme points from the direction of the normal vector because the change of the intensity value in the direction of the normal vector is the largest. The normal vector can be calculated by the Hessian matrix using the method introduced in Section 2.3.1. Let $\boldsymbol{u} = \left( \delta n_x, \delta n_y \right)^T$ represent the normal vector and $\delta$ represent the coefficient that controls the norm. Then the partial derivative of the pair can be obtained as follows:

$$\frac{\partial I_f \left( \delta n_x, \delta n_y \right)}{\partial \delta} = n_x r_x + n_y r_y + \delta n_x^2 r_{xx} + 2\delta n_x n_y r_{xy} + \delta n_y^2 r_{yy} \tag{11}$$

To make the intensity value can obtain the extreme point in the direction of the normal vector, the value of Equation (11) should be set to 0. The magnitude of the coefficient $\delta$ can be obtained by using Equation (12), and the relative offset can also be obtained by Equation (13). Since the pixels are closely connected, the adjacent pixels need to divide the range equally, so the offset at this point is only valid when $(\Delta x, \Delta y) \in \left[-\frac{1}{2}, \frac{1}{2}\right] \times \left[-\frac{1}{2}, \frac{1}{2}\right]$ is specified. The sub-pixel coordinate of the center point can be obtained by adding the offset to the original coordinate point according to Equation (14). Exactly, suppose that we select a candidate point of an interference ring in the two-dimensional spectrum and map the intensity value of its normal vector direction to one-dimensional space for analysis. As shown in Figure 7, the light intensity profile is approximately a Gaussian curve, and the position of the extreme point of the second-order Taylor expansion is closer to the real centerline than the candidate point. Figure 8 shows the spatial distribution of the original candidate points (pixel level) of the experimental spectrum after Frangi filtering and the sub-pixel points formed by adding the offset. It can be seen that the sub-pixel points are closer to the centerline than the candidate points.

$$\delta = -\frac{n_x r_x + n_y r_y}{n_x^2 r_{xx} + 2n_x n_y r_{xy} + n_y^2 r_{yy}} \tag{12}$$

$$(\Delta x, \Delta y) = (\delta n_x, \delta n_y) \tag{13}$$

$$p_e = (x_o + \Delta x, y_o + \Delta y) \tag{14}$$

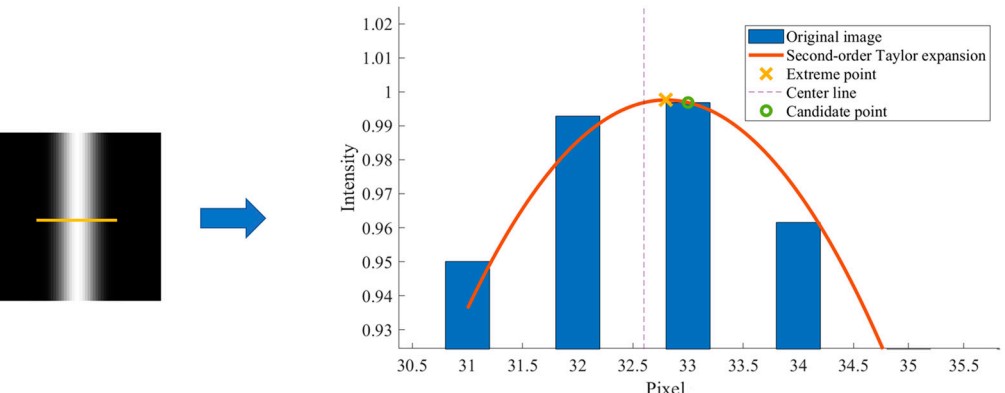

**Figure 7.** Intensity value in the direction of the normal vector of the interference ring.

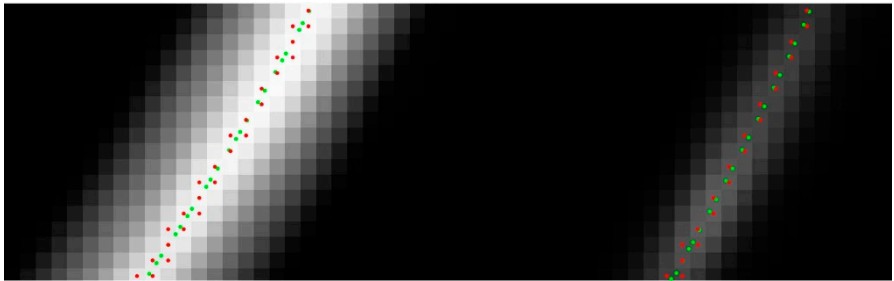

**Figure 8.** The location result of the center points of the interference ring in the experimental spectrum. (Red points are the candidate points, and green points are the final sub-pixel points.)

### 2.3.3. Curve Fitting

After the center points are identified based on the above process, the 8-neighbor connection algorithm is used to improve the execution speed of the algorithm. The param-

eters $a$, $b$, and $r$ of the interference ring can be fitted by using Equation (15) and the least squares method.

$$(x - a)^2 + (y - b)^2 = r^2 \qquad (15)$$

To verify the influence of noise and missing signal parts on the curve fitting error, four spectra with different noises were simulated with the resolution of 1024 × 256 pixels (The noise level is the same as that of the experimental spectra), as shown in Figure 9. The first, second, and third level rings with larger radii are selected (provided the innermost level is level 0) as the object of curve fitting. The processing results are shown in Table 2. It can be seen that the fitted circle center coordinates do not exceed 0.25 pixels. Most importantly, the deviation of the radius does not exceed 0.1 pixels, and the corresponding frequency shift error <10 MHz, which proves that our method presents high accuracy for extracting the frequency shift of Brillouin scattering.

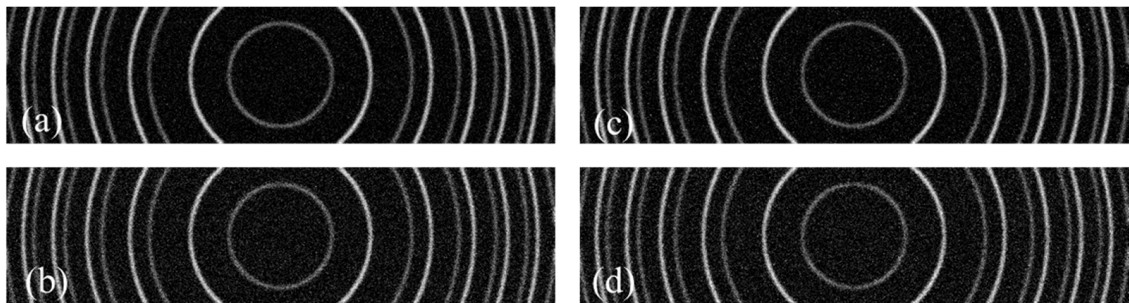

**Figure 9.** Simulated spectra with different noises. (**a**) Gaussian noise 0.025 + salt and pepper noise 0.001; (**b**) Gaussian noise 0.05 + salt and pepper noise 0.001; (**c**) Gaussian noise 0.025 + salt and pepper noise 0.005; and (**d**) Gaussian noise 0.05 + salt and pepper noise 0.005.

**Table 2.** Processing results of simulated spectra after adding different noises.

| Simulated Spectra | Noise Type | $\Delta a$ (Pixel) | $\Delta b$ (Pixel) | $\Delta r$ (Pixel) |
|---|---|---|---|---|
| (a) | Gaussian noise 0.025 + salt and pepper noise 0.001 | 0.02 | 0.11 | 0.03 |
| (b) | Gaussian noise 0.05 + salt and pepper noise 0.001 | 0.04 | 0.25 | 0.07 |
| (c) | Gaussian noise 0.025 + salt and pepper noise 0.005 | 0.03 | 0.08 | 0.04 |
| (d) | Gaussian noise 0.05 + salt and pepper noise 0.005 | 0.04 | 0.19 | 0.07 |

## 3. Results

The measurements to invert the seawater temperature was conducted by using the SBS lidar system. According to the annual average temperature and salinity distributions of the upper-ocean mixed layer [25,26], pure water and seawater with a salinity of 30‰ were prepared by dissolving sea salt (Sigma-Aldrich) in distilled water. The water temperature was stabilized to values between 10 and 30 °C. Twenty spectra were collected for each set of experimental measurements at the same temperature. The measured spectra were processed using the method proposed in Section 2. In the curve fitting stage, we classify each connected line (represented by different colors) according to the signal category and then complete the curve fitting separately. Judging from the partially enlarged spectrum in Figure 10 of the processing result, the proposed algorithm can accurately locate the position of the ring center. It can also be seen that the fitted circle and the actual interference ring are highly fitted.

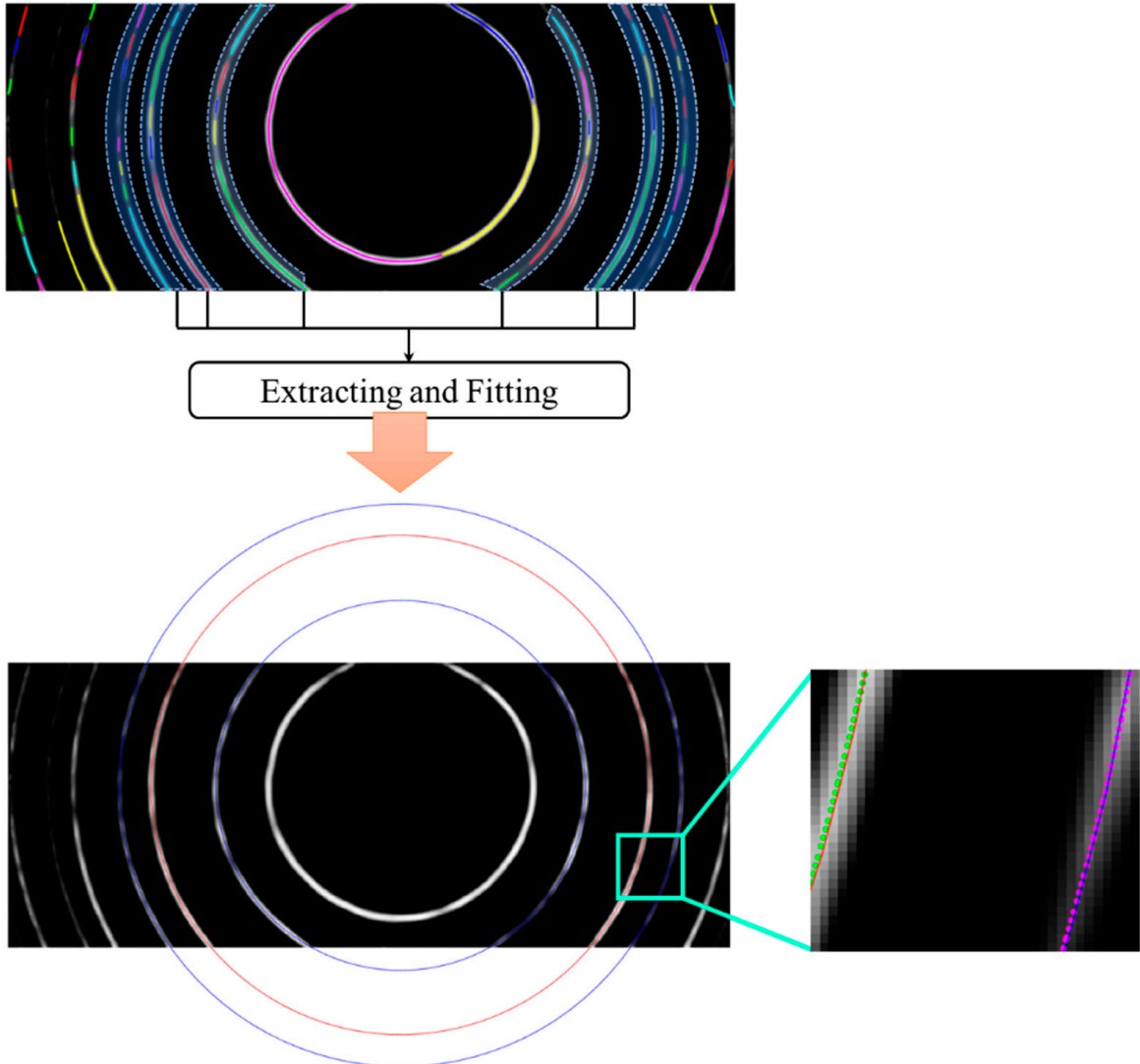

**Figure 10.** Spectrum processing of the experimental measurement.

Based on the obtained radius values of interference rings, the frequency shift of Brillouin scattering can be calculated, and the seawater temperature can be obtained by combining the fitting formula [18]. Figure 11a shows the measured results. The deviations of frequency shift and temperature are shown in Figure 11b,c, respectively. The frequency shift deviation value is the difference between the measured value and the fitting formula, and the temperature deviation value is the difference between the inversion temperature and the temperature measured by the thermocouple instrument.

As can be seen from Figure 11b,c, most of the measured errors of frequency shift are up to an order of MHz. The minimum error of frequency shift can be as low as 2 MHz, and the minimum error of inversion temperature is 0.14 °C (including the uncertainty of the fitting formula in the inversion). Therefore, the accuracy of our proposed method is quite impressive when dealing with the spectra of Brillouin lidar for measuring seawater temperature. More specific investigations on the automatic recognition and classification method based on the artificial intelligence algorithm to improve the performances of SBS lidar will be the subject of further studies.

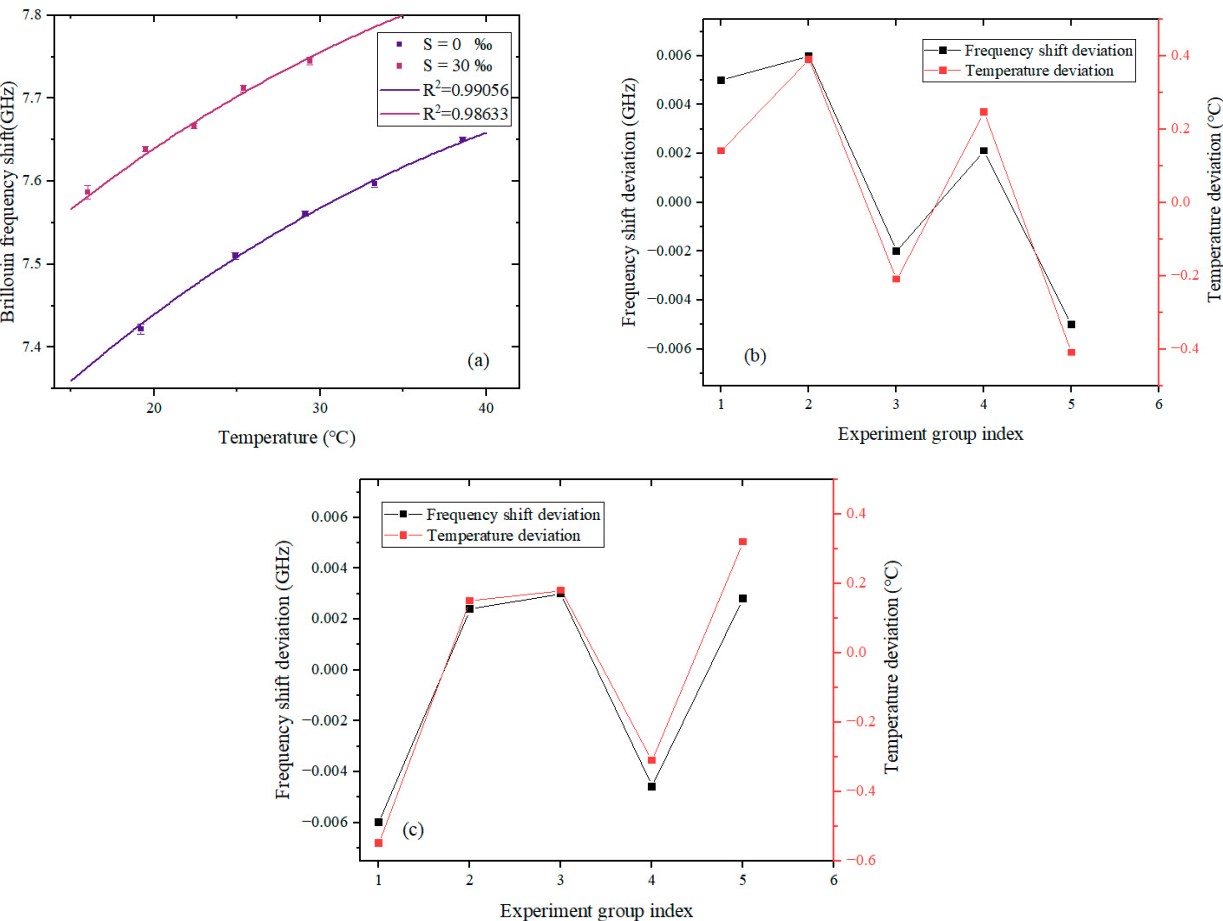

**Figure 11.** The experimental results of temperature measurement. (**a**) calculated frequency shift versus the water temperature at different salinities; (**b**) frequency shift and temperature deviations of water with a salinity of 30‰; (**c**) frequency shift and temperature deviations with a salinity of 0‰.

## 4. Discussion

In the collected spectra by the interferometer system, there are multiple interference level rings. Because they all meet the requirements of Equation (1), there are several options for calculating Brillouin frequency shift. In our processing process, we also compared the calculated results of the frequency shift of Brillouin scattering by using different options. In our experimental spectra, the interference rings selected for the calculation can be divided into two types: the innermost available (expressed as "Inner") and the adjacent level of Inner (expressed as "Outer"). The inner and outer rings are shown in Figure 12. We conducted a group of experiments in pure water with a temperature of 25.4 °C, the results of which are shown in Table 3. Furthermore, the proposed method is compared with the data fold method [15] and the cylindrical lens compression method [14], respectively. It can be seen that the result has smaller errors by using inner rings compared to using outer rings. Therefore, when inner rings are available (with regular shape and brightness exceeding the lowest value recognized by the algorithm), they will be given priority as the object of the frequency shift calculation of Brillouin scattering. Moreover, the proposed method in this paper has less deviation and uncertainty than that of the data fold and cylindrical lens compression methods.

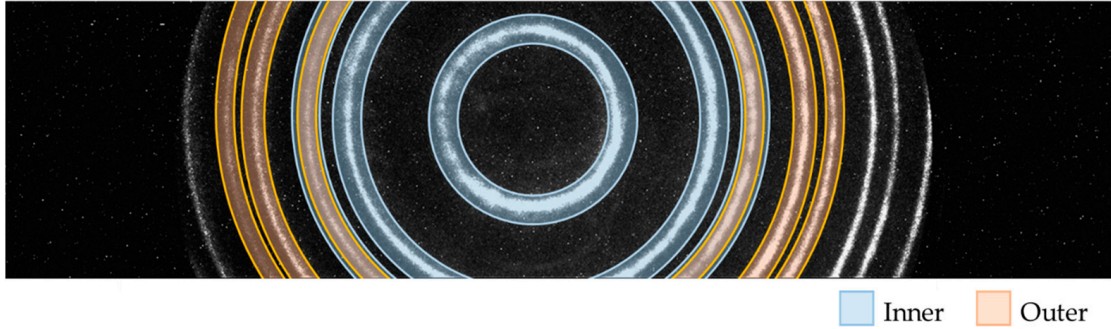

**Figure 12.** Distribution diagram of inner and outer rings.

**Table 3.** Comparison of experimental results using different methods.

| Method | Option | Average Frequency Shift Deviation (MHz) | Average Measurement Uncertainty (MHz) |
|---|---|---|---|
| Proposed | | 3.12 | 4.60 |
| Data fold | Inner | 12.88 | 6.36 |
| Cylindrical lens compression | | 9.13 | 8.84 |
| Proposed | | 3.96 | 7.80 |
| Data fold | Outer | 14.20 | 16.66 |
| Cylindrical lens compression | | 16.37 | 10.99 |

## 5. Conclusions

In summary, we propose a high-accuracy spectral measurement method of SBS lidar based on the Hessian matrix and Steger algorithm for improving the measurement accuracy of seawater temperature. Firstly, the Frangi filter based on the Hessian matrix is used to enhance the curve structure while suppressing noise and reducing random errors. Then the Steger algorithm is used to extract the center point of the interference rings to accurately locate the position of the interference rings. Finally, the position coordinates of the center point of the interference rings are regarded as the observed values, and the equation is fitted by the least square method to obtain the center coordinates and the radii of the interference rings. Based on the obtained radii, the frequency shift of Brillouin scattering has been calculated. The results show that most of the measured errors of frequency shift are generally at an order of MHz, and the accuracy of the inversion temperature can be as low as 0.14°C. In terms of operational complexity, compared with the related processing methods proposed before, the methods proposed in this paper only require a small amount of manual annotation, which greatly simplifies the operation steps. The results of our study will be essential to Brillouin lidar in remote sensing of seawater parameters in the ocean.

**Author Contributions:** Conceptualization and methodology, J.S. (Jiulin Shi); validation, Z.L., J.S. (Jie Sun), X.Z., Z.Z. and Y.X.; formal analysis, N.L. and X.H.; investigation, Z.L.; writing—original draft preparation, Z.L.; writing—review and editing, J.S. (Jiulin Shi); supervision, J.S. (Jiulin Shi); project administration, J.S. (Jiulin Shi); funding acquisition, J.S. (Jiulin Shi). All authors have read and agreed to the published version of the manuscript.

**Funding:** This research was funded by the National Natural Science Foundation of China (41776111) and the Defense Industrial Technology Development Program (JCKY2019401D002).

**Data Availability Statement:** Data underlying the results presented in this paper are not publicly available at this time but may be obtained from the authors upon reasonable request.

**Acknowledgments:** We thank the reviewers for their careful reading and valuable comments, which helped us to improve the manuscript.

**Conflicts of Interest:** The authors declare no conflict of interest.

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
