# Peer review of "High-Accuracy Spectral Measurement of Stimulated-Brillouin-Scattering Lidar Based on Hessian Matrix and Steger Algorithm"

_remotesensing, doi:10.3390/rs15061511_

Round 1

Reviewer 1 Report

In this paper, the author proposed an image processing algorithm to deal with the stimulated Brillouin scattering spectrum. The author used Hessian matrix and Frangi filter to denoise the RBS image produced from ICCD and used Steger algorithm to determine the center points of scattering spectra. The Brillouin shift is finally calculated by using the center and radii of

ring interference spectrum and used for temperature retrieving. The results show that the inversion accuracy of temperature can be as low as 0.14℃. This is a very useful working for the area of RBS but the paper needs improvement before accepted for publication. My detailed comments are as follows:

 1. Why does the Gaussian form (Eq.(3) ) can be used to determine the first and second order partial derivatives in Hessian matrix?

 2. In Table 2, the ‘â–³C’ should be ‘â–³r’.

 3. For figure 9, the author adds Gaussian noise and salt and pepper noise with different levels. What is the basis for choosing the level of the nosies?

 4. Generally, the passion noise will also be produced in ICCD and what is its impact on the results?

 5. In line 306, the Fig.10 should be Fig.11 including subsequent writing.

 6. The format of references should be consistent.

Author Response

Thank you very much for your positive comments and suggestions! We have revised the manuscript according to your comments and suggestions, and we hope that the revised manuscript is now suitable for publication in Remote Sensing.

Comment 1: Why does the Gaussian form (Eq.(3) ) can be used to determine the first and second order partial derivatives in Hessian matrix?

Response: Thank you very much! By solving Eq (3), the first and second partial derivatives can be used to obtain the convolution kernels of the first and second order partial derivatives by restricting the values of x and y as integers within the range of  . The first and second order partial derivatives of the Hessian matrix can be obtained by performing the convolution operation with the original image one by one. There are many formulas to calculate the convolution kernel. The purpose of choosing Gaussian formula is to suppress noise as much as possible and obtain smooth results.

Comment 2: In Table 2, the ‘â–³C’ should be ‘â–³r’.

Response: Thank you very much! We have revised the symbol in our manuscript.

Comment 3: For figure 9, the author adds Gaussian noise and salt and pepper noise with different levels. What is the basis for choosing the level of the nosies?

Response: Thank you very much! According to the collected spectra in our experiment, we found that the noises are mainly salt and pepper noise and Gaussian noise by using histogram statistics. We have calculated the noise level of these spectra and found that the percentage of salt and pepper noise is in the range of 1‰ - 4‰, and the variance of Gaussian noise is 6-20 (gray value). Therefore, the simulated spectra were added with the same level and more noise to test the accuracy of the method.

Comment 4: Generally, the passion noise will also be produced in ICCD and what is its impact on the results?

Response: Thank you very much! The influence of passive noise is mainly reflected in the randomness of the background light intensity of spectra and some salt and pepper noise. Most of the noise is removed in the filtering stage. In addition, to reduce the random impact of passive noise on the results, at least 20 spectra were processed at each temperature.

Comment 5: In line 306, the Fig.10 should be Fig.11 including subsequent writing.

Response: Thank you very much! We have revised the mistake in our manuscript.

Comment 6: The format of references should be consistent.

Response: Thank you very much! We unified the format of the references.

Reviewer 2 Report

This manuscript proposes a spectral measurement method of SBS Lidar based on Hessian matrix and Steger algorithm for improving the measurement accuracy of seawater temperature. This method requires a small amount of manual annotation and simplifies the operation steps. In my opinion, there are still a few questions need to be considered.

1.    More analyses should be included in the results part to test the algorithm under different water environments.

2.    The temperature inversion accuracy of the method proposed in this manuscript should be compared in detail with other methods mentioned in the introduction.

Author Response

Thank you very much for your positive comments and suggestions! We have revised the manuscript according to your suggestions, and we hope that the revised manuscript is now suitable for publication in Remote Sensing.

Comment 1: More analyses should be included in the results part to test the algorithm under different water environments.

Response: Thank you very much! We have added a group of experimental data with the water temperature of 19.2℃, 24.9℃, 29.1℃, 33.3℃, and 38.6℃ to test the algorithm, and the error between forecast and the true value has also been analyzed in the result analysis section.

Comment 2: The temperature inversion accuracy of the method proposed in this manuscript should be compared in detail with other methods mentioned in the introduction.

Response: Thank you very much! We have compared the temperature inversion accuracy of the method proposed in our manuscript with the results reported in other methods.

Reviewer 3 Report

Authors contributions:

The authors have proposed a method for extraction of the frequency, shift with high precision from stimulated Brillouin scattering Lidar, based on Hessian matrix and Steger algorithm.

The following steps are used:

ü  Hessian matrix and Frangi filter are used for signal denoising.

ü  The center points of scattering spectra at the sub-pixel level are extracted by the Steger algorithm, and then the center points are connected and classified according to the signal type.

ü  The temperatures of seawater with the salinity of 30‰ are inverted by using the frequency shifts of Brillouin scattering.

I have some reviewer notes:

Title. If it is possible, add full name of the abbreviation SBS.

Abstract. How your work improves the known solutions in this study area?

Introduction part. At the end of this part, the aim of this work have to be clearly defined.

Figure 5. If Intensity, Pixel abscissa and Pixel ordinate are dimensionless, you have to describe it in the text. Also, the description has to be above it. Same for other figures.

Table 1. The eigenvalues have to be presented as values.

Line 272. Is the resolution 1024×256 pix enough for your analysis.

Line 292. How did you choose this sample size of 20 spectra? It is not clear, are 20 spectra enough.

Figure 11. What is the accuracy of your linear models, R^2, some error values?

Discussion part. In this part, you have to compare your results with minimum three other papers.

Conclusion part. How your work improves the known solutions in this study area?

How your work will be continued?

What are the limitations of the proposed method?

I have some suggestions:

Improve presentation of your results. Make more comparative analyses. These suggestions will improve your contributions.

Author Response

Thank you very much for your positive comments and suggestions! We have revised the manuscript according to your suggestions, and we hope that the revised manuscript is now suitable for publication in Remote Sensing.

Comment 1: Title. If it is possible, add full name of the abbreviation SBS.

Response: Thank you very much! We have added the full name of the abbreviation SBS in the title.

Comment 2: Abstract. How your work improves the known solutions in this study area?

Response: Thank you very much! We have revised the Abstract section. Due to the limited resolution of ICCD camera, the traditional processing methods remain at the level of pixel or partial sub-pixel, which cannot meet the requirements of high-performance Lidar. In this paper, the Steger algorithm with sub-pixel recognition accuracy combined with the least square fitting method breaks through the resolution limit, which improves accuracy of extracting frequency shift significantly.

Comment 3: Introduction part. At the end of this part, the aim of this work has to be clearly defined.

Response: Thank you very much! We have revised the Introduction section.

Comment 4: Figure 5. If Intensity, Pixel abscissa and Pixel ordinate are dimensionless, you have to describe it in the text. Also, the description has to be above it. Same for other figures.

Response: Thank you very much! We have revised the figures in our manuscript.

Comment 5: Table 1. The eigenvalues have to be presented as values.

Response: Thank you very much! We have revised Table 1 in our manuscript.

Comment 6: Line 272. Is the resolution 1024×256 pix enough for your analysis. Response: Thank you very much! The pixel size of ICCD has influence on the error.   

The larger the pixel size the larger the measuring error. The pixel size of the CCD we used is 26 µm, it will result in an extra measuring error of Brillouin shift. Therefore, an ICCD camera with smaller pixel size is preferable for reducing measuring error. An ICCD camera with smaller pixel size will be used to improve the measurement accuracy in our further studies.

Comment 7: Line 292. How did you choose this sample size of 20 spectra? It is not clear, are 20 spectra enough.

Response: Thank you very much! We have selected 20 spectra at each temperature, and the total number of processed images exceeds 200. When the 20 spectra are processed, the change of the mean and variance of the extracted Brillouin frequency shift has been very small, and its inversion has reached a stable value.

Comment 8: Figure 11. What is the accuracy of your linear models, R^2, some error values?

Response: Thank you very much! We have added the accuracy and error values of the linear models in our manuscript.

Comment 9: Discussion part. In this part, you have to compare your results with minimum three other papers.

Response: Thank you very much! We have compared our results with the results obtained by using data folding and cylindrical lens methods in Discussion section.

Comment 10: Conclusion part. How your work improves the known solutions in this study area?

Response: Thank you very much! This paper proposes a novel method to access the spectral information of SBS Lidar. Compared with known solutions in this study area, our method has much higher precision in image pixel processing. In addition, the fitting method is used to find the optimal parameter value, which further improves the accuracy of extracting Brillouin frequency shift. On the other hand, the spectral processing process has also been greatly simplified.

Comment 11: How your work will be continued?

Response: Thank you very much! We will consider adding automatic classification function, which can select the required center line segment according to the signal type for fitting, so as to achieve the purpose of fully automatic processing of SBS lidar spectral image.

Comment 12: What are the limitations of the proposed method?

Response: Thank you very much! The main limitation is that if the interference ring in the interference image is missing and asymmetric, the accuracy of extracting Brillouin frequency shift will be damaged.

Comment 13: I have some suggestions: Improve presentation of your results. Make more comparative analyses. These suggestions will improve your contributions.

Response: Thank you very much! We have added the comparative analyses with the known methods in our manuscript.

Round 2

Reviewer 2 Report

The comments are well answered.